# Crystal Structure of DNA Replication Protein SsbA Complexed with the Anticancer Drug 5-Fluorouracil

**DOI:** 10.3390/ijms241914899

**Published:** 2023-10-04

**Authors:** Hsin-Hui Su, Yen-Hua Huang, Yi Lien, Po-Chun Yang, Cheng-Yang Huang

**Affiliations:** 1Department of Pharmacy, Chia Nan University of Pharmacy and Science, Tainan City 717, Taiwan; 2Department of Biomedical Sciences, Chung Shan Medical University, Taichung City 402, Taiwan; 3Department of Biological Sciences, Purdue University, West Lafayette, IN 47907, USA; 4Department of Medical Research, Chung Shan Medical University Hospital, Taichung City 402, Taiwan

**Keywords:** 5-fluorouracil, SSB, anticancer drug, single-stranded DNA-binding protein, SsbA, crystal structure, docking, conformational change

## Abstract

Single-stranded DNA-binding proteins (SSBs) play a crucial role in DNA metabolism by binding and stabilizing single-stranded DNA (ssDNA) intermediates. Through their multifaceted roles in DNA replication, recombination, repair, replication restart, and other cellular processes, SSB emerges as a central player in maintaining genomic integrity. These attributes collectively position SSBs as essential guardians of genomic integrity, establishing interactions with an array of distinct proteins. Unlike *Escherichia coli*, which contains only one type of SSB, some bacteria have two paralogous SSBs, referred to as SsbA and SsbB. In this study, we identified *Staphylococcus aureus* SsbA (SaSsbA) as a fresh addition to the roster of the anticancer drug 5-fluorouracil (5-FU) binding proteins, thereby expanding the ambit of the 5-FU interactome to encompass this DNA replication protein. To investigate the binding mode, we solved the complexed crystal structure with 5-FU at 2.3 Å (PDB ID 7YM1). The structure of glycerol-bound SaSsbA was also determined at 1.8 Å (PDB ID 8GW5). The interaction between 5-FU and SaSsbA was found to involve R18, P21, V52, F54, Q78, R80, E94, and V96. Based on the collective results from mutational and structural analyses, it became evident that SaSsbA’s mode of binding with 5-FU diverges from that of SaSsbB. This complexed structure also holds the potential to furnish valuable comprehension regarding how 5-FU might bind to and impede analogous proteins in humans, particularly within cancer-related signaling pathways. Leveraging the information furnished by the glycerol and 5-FU binding sites, the complexed structures of SaSsbA bring to the forefront the potential viability of several interactive residues as potential targets for therapeutic interventions aimed at curtailing SaSsbA activity. Acknowledging the capacity of microbiota to influence the host’s response to 5-FU, there emerges a pressing need for further research to revisit the roles that bacterial and human SSBs play in the realm of anticancer therapy.

## 1. Introduction

Nucleobases play a pivotal role as fundamental constituents within nucleic acids, orchestrating the replication of genetic information across all biological systems [1]. The accurate synthesis of nucleotides stands as a critical linchpin for the survival and proliferation of both eukaryotic and prokaryotic cells [2]. The structural alteration of nucleobases holds the potential to exert substantial impacts, displaying potent biological effects [3]. Throughout an expansive range encompassing anticancer [4], antiviral [5], antibacterial [6], anti-inflammatory [7], and antitumor activities [8], numerous derivatives of uracil have garnered longstanding employment. One standout exemplar in this domain is the FDA-approved anticancer agent, 5-fluorouracil (5-FU) [4]. In 5-FU, the hydrogen situated at the C5 position of uracil is supplanted by a fluorine atom, culminating in a fluoropyrimidine configuration. This modification empowers 5-FU to effectively target the enzyme thymidylate synthase (TSase) for anticancer chemotherapy [9]. The cytotoxic influences of 5-FU emerge through its adept ability to impede the operation of TSase, induce RNA miscoding, and activate apoptosis. In the dynamic landscape of drug development, although numerous novel agents have been conceived, 5-FU persists as a cornerstone within the arsenal of chemotherapeutic modalities. It prominently features in systemic treatments for an array of cancers spanning the gastrointestinal tract, breast, head, and neck [9]. Notably, 5-FU manifests diverse interactions with over a dozen distinct proteins, including dihydropyrimidinase (DHPase) [10], an enzyme involved in the pyrimidine degradation [11]. Strikingly, 5-FU-induced toxicity has been found in asymptomatic patients with DHPase deficiency. Such patients, undergoing anticancer therapy with 5-FU, confront severe toxicity manifestations, including instances of mortality [12]. Beyond the confines of human genetics [12,13] and the associations with human gene products [14], the intricate web of host–microbiota interactions ushers in additional dimensions to the 5-FU narrative [15,16,17]. Evidently, the active gut microbiota, equipped with the capacity to synthesize bromovinyluracil, can exert profound regulatory influences on systemic 5-FU concentrations [17,18]. This unforeseen modulation results in an adverse outcome, as demonstrated by a tragic occurrence where 5-FU exposure triggered the demise of 16 patients in Japan [17,18]. Given the intricate tapestry of interactions that envelop 5-FU, the imperative emerges to comprehensively construct its interactome. Such a feat is essential for in-depth analyses of clinical pharmacokinetics and toxicity profiles [19,20,21,22]. The holistic elucidation of the multifaceted relationships woven by 5-FU holds the promise of enhancing our grasp of its intricate dynamics, paving the way for refined therapeutic strategies and personalized medicine. To attain this objective, the initial stride involves the identification of additional proteins that interact with 5-FU.

DNA is subject to constant challenges, such as replication errors, DNA damage, and the formation of secondary structures [23]. To ensure its integrity and stability, cells have evolved an intricate network of proteins involved in DNA metabolism [24,25,26]. Single-stranded DNA (ssDNA)-binding protein (SSB) is one of central players in these processes, safeguarding and manipulating ssDNA during various cellular events [27]. Through its multifaceted roles in DNA replication, recombination, repair, replication restart, and other cellular processes, SSB emerges as a critical player in maintaining genomic integrity [28,29,30]. SSB binds specifically to ssDNA with high affinity, preventing its re-annealing, protecting it from nucleases, and promoting its accessibility to other DNA-binding proteins [31]. SSBs are conserved across organisms, from bacteria [32] to humans [33,34], highlighting their fundamental importance. SSBs have undergone extensive investigation in eubacteria, with a notable focus on the *Escherichia coli* SSB (EcSSB) [31]. The majority of SSBs adopt homotetrameric configurations for activity [35]. This arrangement involves four oligonucleotide/oligosaccharide-binding folds (OB folds) coalescing to constitute a DNA-binding domain [36,37]. Beyond their DNA-binding functions, SSBs also establish interactions with a multitude of DNA-metabolism proteins, collectively forming the SSB interactome [38]. EcSSB comprises two primary domains: an N-terminal ssDNA-binding/oligomerization domain (SSBn) and a flexible C-terminal domain for protein–protein interactions (SSBc). Within EcSSBc, a further division into two sub-domains emerges, namely an intrinsically disordered linker (IDL) and an acidic tip [39]. Bacterial SSBs from distinct sources share moderate sequence homology, especially within the SSBn domain, encompassing roughly the initial 110 residues. The ubiquity of this homology designates SSBn as a prospective common target for devising inhibitors against various SSBs [40,41,42,43]. Diverging from *E. coli*, which harbors a solitary SSB variant (EcSSB), certain bacteria like *Staphylococcus aureus* exhibit a more complex scenario with the presence of three paralogous SSBs, specifically referred to as SaSsbA [44,45], SaSsbB [46], and SaSsbC [47]. Of these, SaSsbA may be an EcSSB counterpart due to its possession of an acidic tip and sequence resemblance to EcSSBn. Given this alignment, the ongoing pursuit of molecules that bind to and inhibit SaSsbA holds notable promise for future applications in combating pathogens [44,48]. Although the identification of SaSsbB as a protein that binds to 5-FU is established [49], it holds significance to investigate whether 5-FU can also interact with SaSsbA, and delving into the mode of this interaction is a valuable pursuit.

Although 5-FU has been under investigation for over six decades [50,51], its interactions with proteins remain challenging to predict. In this study, we identified SaSsbA as a fresh addition to the roster of 5-FU binding proteins, and, thus, the 5-FU interactome was extended to include this essential DNA replication protein. In our pursuit of comprehending the precise interaction sites between 5-FU and SaSsbA, we solved the complexed crystal structure at a resolution of 2.3 Å (PDB ID 7YM1). The structure of SaSsbA complexed with glycerol was also determined at 1.8 Å (PDB ID 8GW5). The binding sites within the structure of this OB-fold protein emerge as prime contenders for potential drug design efforts. To further characterize the binding affinity and confirm the interacting sites of 5-FU with SaSsbA, fluorescence quenching and mutational analysis were conducted. In addition, we also compared the 5-FU binding modes between SaSsbA and SaSsbB. The interaction of SaSsbB with 5-FU relies on specific residues T12, K13, T30, F48, and N50 [49], and these residues remain conserved within SaSsbA [45,46]. Given that SaSsbA exhibits structural parallels with SaSsbB, one might initially infer that 5-FU’s binding capabilities would align, and the 5-FU binding mode of SaSsbA would echo that of SaSsbB. However, our investigation unveiled a disparity between their 5-FU binding sites. Accordingly, this complexed crystal structure unfurls a molecular insight into the distinct manner in which the anticancer drug 5-FU binds to a cognate protein, even when the structural scaffold appears analogous yet divergent.

## 2. Results

### 2.1. Sequence Analysis of SaSsbA

According to the nucleotide sequence available on NCBI, the projected monomeric SaSsbA protein spans 167 amino acid residues (aa), with a calculated molecular mass of 19 kDa. The alignment consensus of sequenced SSB homologs, facilitated by ConSurf analysis, delineated the extent of variability at distinct positions along the sequence (Figure 1). It became apparent that the aa 107–148 segment (teal) in SaSsbA lacks conservation within the SSB homologs. Drawing from insights garnered from the EcSSB-ssDNA complex [52], it emerges that four crucial aromatic residues, namely W40, W54, F60, and W88, universally preserved within most SSB families as F/Y/W, engage in ssDNA binding through stacking interactions. Correspondingly, in SaSsbA, the corresponding residues manifest as F37, F48, F54, and Y82. A notable absence in SaSsbA is the W residue. Moreover, the EcSSB’s significant C-terminal tail DDDIPF, which plays a role in protein–protein interaction, undergoes a modification to DDDLPF in SaSsbA. The GGRQ motif postulated as a regulatory switch for ssDNA binding [53], which in SaSsbA could be replaced by the GGQR motif (aa 112–115). In the context of the PXXP motifs within EcSSB, positioned at aa 139 (PQQP), 156 (PQQS), and 161 (PAAP), and acknowledged for their role in mediating protein–protein interactions [39], their presence is notably absent in SaSsbA (Figure 1). While SaSsbA is presumed to be a primary SSB homolog similar to EcSSB in structure and function, divergences arise in their gene locations [45]. In the genetic map of *S. aureus*, the *ssbA* gene resides flanked by *rpsF* (encoding the ribosomal protein S6) and *rpsR* (encoding ribosomal protein S18) genes, all encompassed within one operon regulated by the SOS response [54]. The scenario deviates in *E. coli*, where the *ssb* gene is found adjacent to the *uvrA* gene, positioned distant from that *S. aureus* operon. Instead, the *prib* gene (coding for PriB, an ssDNA-binding protein) [55] is flanked by the *rpsF* and *rpsR* genes in the genetic map of *E. coli*. The rationale behind the necessity for the evolution of distinct SSBs in particular species remains a subject that warrants elucidation.

### 2.2. Crystallization of the Glycerol-Bound SaSsbA and the SaSsbA-5-FU Complex

SaSsbA with a His tag was overexpressed in *E. coli* through heterologous expression and subsequently isolated from the soluble supernatant using Ni^2+^-affinity chromatography. The purified SaSsbA was concentrated to a concentration of 20 mg/mL, with the addition of the cryoprotectant glycerol to attain a final concentration of 25%. This glycerol-enhanced formulation allowed for storage at −20 °C. Previously, we observed that crystals of apo-SaSsbA could be cultivated at room temperature using the hanging drop vapor diffusion method in a solution composed of 22% PEG 4000, 100 mM HEPES, and 100 mM sodium acetate at pH 7.5 [45]. Efforts were initially directed towards soaking and cocrystallization of SaSsbA (20 mg/mL) with 5-FU (200 μM) under identical crystallization conditions to those for apo-SaSsbA. The goal was to obtain crystals of the SaSsbA–5-FU complex, yet these attempts proved unfruitful. Subsequent rescreening was undertaken utilizing commercial crystallization kits. Under these new conditions (JBScreen Classic 2, Jena Bioscience, Jena, Germany), crystals of the SaSsbA–5-FU complex emerged at room temperature within a mixture containing 16% PEG 4000, 100 mM Tris-HCl, and 200 mM MgCl_2_ at pH 8.5. For glycerol-bound SaSsbA, crystallization occurred in a solution containing 30% PEG 4000, 100 mM HEPES, and 200 mM CaCl_2_ at pH 7.5 (Table 1).

### 2.3. Crystal Structure of Glycerol-Bound SaSsbA

The crystal structure of glycerol-bound SaSsbA was successfully determined at a resolution of 1.8 Å (Table 1). The crystals of glycerol-bound SaSsbA belonged to the P4_1_2_1_2 space group, featuring cell dimensions with *a* = 88.22 Å, *b* = 88.22 Å, and *c* = 58.00 Å. This structure of SaSsbA with glycerol (PDB ID 8GW5) was elucidated via molecular replacement, utilizing the apo-SaSsbA as a model (PDB ID 5XGT). Examination of a Ramachandran plot revealed no presence of unallowed regions (outliers) within this structure. While SaSsbA is typically active as tetramers [45], this glycerol-bound structure contained only two monomers of SaSsbA within each asymmetric unit (Figure 2A). In contrast to the crystal structure of apo-SaSsbA [45], this structure displayed two additional amino acid residues, namely P105 and K106, as indicated by the blue mesh (Figure 2A), solely in the subunit A of SaSsbA. The C-terminal region in SaSsbA, comprising aa 107–167 in the subunit A and aa 105–167 in the subunit B, was not observed. This suggests that the C-terminal region of SaSsbA exhibits dynamic behavior, a feature reminiscent of EcSSB [56]. For this SaSsbA dimer, the electron density mostly exhibited satisfactory quality. Nonetheless, some sections remained disordered and unobserved, including aa 38–44 (loop L_23_) in subunit A and aa 40–42 (loop L_23_) in subunit B within the ternary structure of this glycerol-bound SaSsbA. In congruence with the apo form, the overall architecture of this glycerol-bound SaSsbA monomer remained characteristic of an OB-fold structure, marked by a β-barrel (composed of 5 β-strands) crowned with an α-helix. It is noteworthy that SaSsbA did not encompass the β6 strand, a component found in numerous other SSBs such as those from *E. coli* [52], *Salmonella enterica* [41], *Klebsiella pneumoniae* [53], and *Pseudomonas aeruginosa* [40,42,46,57,58]. In tetrameric SSBs, the β6 strand has been proposed to be involved in mediating diverse protein–DNA and protein–protein interaction specificities among distinct SSBs [59]. Within this structure of an SaSsbA dimer, two glycerol molecules (designated as Glycerol 1 and Glycerol 2) were present. However, these glycerol molecules exhibited distinct binding patterns to SaSsbA (see below).

### 2.4. Crystal Structure of SaSsbA Complexed with 5-FU

The complexed crystal structure of SaSsbA with 5-FU (PDB ID 7YM1) was successfully determined at a resolution of 2.3 Å (Table 1) with molecular replacement employing the apo-SaSsbA as a model (PDB ID 5XGT). The crystals of the SaSsbA–5-FU complex were categorized under the P4_1_2_1_2 space group, showcasing cell dimensions of *a* = 88.09 Å, *b* = 88.09 Å, and *c* = 57.78 Å. The completeness exceeded 99%. Upon scrutinizing a Ramachandran plot, there were no regions featuring unallowed conformations (outliers) within this structure. The electron density mostly exhibited satisfactory quality. However, aa 39–44 (loop L_23_) in subunit A and aa 39–41 (loop L_23_) in subunit B within the ternary structure of the SaSsbA–5-FU complex were disordered and unobserved. In both subunits, the range of aa 105–167 was also not detected. The individual monomer within this complexed SaSsbA revealed the characteristic OB-fold structure, featuring a β-barrel comprising 5 β-strands capped with an α-helix. Within this complex structure of a SaSsbA dimer, a single glycerol molecule and one 5-FU molecule were encapsulated (Figure 2B). The positioning of this glycerol molecule, designated as Glycerol 3, within the SaSsbA–5-FU complex closely mirrored that of Glycerol 1 within the glycerol-bound SaSsbA.

### 2.5. Glycerol 1 Binding Mode of SaSsbA

The presence of the cryoprotectant glycerol within the protein solution led to its binding to SaSsbA. In this study, three distinct glycerol binding sites within our two structures were found. Notably, the binding sites for Glycerol 1 and Glycerol 3 exhibited similarity. As revealed by the crystal structure (PDB ID 8GW5), Glycerol 1 is sandwiched by SaSsbA monomers A and B (Figure 2A and Figure 3A). A comprehensive analysis of the interactions transpiring between Glycerol 1 and SaSsbA was undertaken, resulting in the identification of multiple residues that came within contact distance (<4 Å) with the glycerol molecule. Among these interacting residues were F48 (Subunit B), N50 (Subunit B), S79 (Subunit A), R80 (Subunit A), F91 (Subunit A), V92 (Subunit A), and T93 (Subunit A). Delving into the nature of these interactions, hydrogen bonds formed between the ligand and SaSsbA were meticulously examined by leveraging PLIP (the protein–ligand interaction profiler) [60]. In light of the interactions identified through PLIP, it was established that the main chains of R80 and V92, along with the side chains of S79 and T93, partook in hydrogen bonding with Glycerol 1 (Figure 3B).

### 2.6. Glycerol 2 Binding Mode of SaSsbA

Similar to Glycerol 1, which finds itself ensconced between SaSsbA monomers A and B, Glycerol 2 also forms interactions with both monomers (Figure 3C). Nonetheless, it is important to note that their binding poses and spatial arrangements between Glycerol 1 and 2 differed significantly. The interacting residues associated with Glycerol 1 and Glycerol 2 displayed complete variation (PDB ID 8GW5). Specifically, M1 (Subunit A), L2 (Subunit A), N3 (Subunit A), R4 (Subunit A), T36 (Subunit B), F37 (Subunit B), R76 (Subunit A), and D98 (Subunit A) were observed to participate within contact distance (<4 Å) in binding interactions with Glycerol 2 (Figure 3D). A notable aspect of Glycerol 2 binding was the inclusion of a water molecule in the interaction. This water molecule, in conjunction with M1 (Subunit A) and T36 (Subunit B), also contributed to interactions with 5-FU, facilitated through a hydrogen bonding network. The structural evaluation conducted via PLIP underscored that the main chains of M1, N3, and T36, as well as the side chain of R76, were pivotal components in the hydrogen bonding network, fostering the binding of Glycerol 2 (Figure 3D).

### 2.7. Glycerol 3 Binding Mode of SaSsbA

The binding configuration of Glycerol 3 (PDB ID 7YM1) exhibited similarities to that of Glycerol 1 (Figure 3E). In a manner akin to Glycerol 1, Glycerol 3 was positioned between SaSsbA monomers A and B. The interaction between Glycerol 3 and SaSsbA (<4 Å) involved F48 (Subunit B), N50 (Subunit B), S79 (Subunit A), R80 (Subunit A), F91 (Subunit A), V92 (Subunit A), and T93 (Subunit A). The structural evaluation conducted via PLIP revealed that the main chains of R80 and V92, along with the side chain of T93, partook in hydrogen bonding with Glycerol 3 (Figure 3F).

### 2.8. 5-FU Binding Mode of SaSsbA

SsbA, a crucial DNA replication protein, plays multifaceted roles in nucleic acid metabolism [45,54,61,62]. Prior to this study, it remained uncertain whether the FDA-approved clinical drug 5-FU [4], renowned as a prominent pyrimidine derivative in anticancer therapy, could indeed interact with SsbA. Consequently, the complexed crystal structure of SaSsbA with 5-FU was meticulously established to pinpoint the binding site and delve into the binding mechanism (Figure 4A). The electron density corresponding to 5-FU exhibited a well-defined clarity (Figure 4A). The arrangement of 5-FU was discernible, notably due to the positioning of its substituent (Figure 4A). A comprehensive scrutiny was carried out to decipher the interactions between 5-FU and SaSsbA (Figure 4B). Residues R18, P21, V52, F54, Q78, R80, E94, and V96, positioned within a contact distance of <4 Å, were instrumental in the binding of 5-FU. Through analysis conducted using PLIP [60], it was revealed that a water molecule also participated in the binding of 5-FU, facilitated by E94 in SaSsbA, which engaged in water-molecule-mediated hydrogen bonding (Figure 4B). Based on interactions discerned via PLIP, the four side chains of R18, Q78, R80, and E94 were observed to form hydrogen bonds with 5-FU (Figure 3B). The electrostatic potential surface of the SaSsbA complexed with 5-FU unveiled that 5-FU effectively occupied the groove within SaSsbA (Figure 4C), a site significant for single-stranded DNA binding (Figure 4D). The positive (blue) and negative (red) charge distributions underscored that several critical basic residues on the SaSsbA surface, which are exposed to the solvent and collectively form a binding pathway conducive for accommodating ssDNA binding ssDNA (gold). This complex structure insightfully revealed that the presence of 5-FU in the groove potentially influences the wrapping of ssDNA by SaSsbA.

### 2.9. Comparative Analysis of 5-FU Binding Sites in Different Binding States of SaSsbA

In this study, we identified that residues R18, P21, V52, F54, Q78, R80, E94, and V96 within SaSsbA engage in interactions with 5-FU. Furthermore, upon comparing the complexed structures of SaSsbA monomers A and B (PDB ID 7YM1), distinct binding site configurations emerged between the 5-FU-bound state (monomer A) and the Glycerol 3-bound state (monomer B). This comparison revealed noteworthy conformational changes (Figure 5A). When accommodating 5-FU (Figure 5B), the positions of R18 and P21 experienced shifts of approximately 5.2 and 7.3 Å, respectively. Moreover, V52, F54, Q78, and E94 exhibited angular shifts of 180, 22, 23, and 52°, respectively, due to 5-FU binding. In addition, the sizes of the binding groove differed between monomer A (the 5-FU-bound state) and monomer B (the Glycerol 3-bound state) (Figure 5C). While the OB folds exhibited a similar appearance, it was observed that their sizes were divergent. Structurally, the angles between strands β1′ and β4 in monomer A and B were measured at 41.2° and 65.9°, respectively (Figure 5C).

### 2.10. Comparative Structural Analysis of 5-FU Binding Sites in the 5-FU-Bound and Unbound States of SaSsbA

Having previously determined the crystal structure of the apo-form of SaSsbA [45], we have a basis for comparing the structural aspects of the 5-FU binding sites between the 5-FU-bound state (PDB ID 7YM1) and the unbound state (PDB ID 5XGT) of SaSsbA. Through superimposition of these structures, R18 in the apo-form SaSsbA was significantly shifted by a distance of 7.2 Å and an angle of 107.3° upon binding of 5-FU (Figure 5D). Accordingly, the side chain of R18 is likely a pivotal element in facilitating 5-FU binding.

### 2.11. Structure-Based Mutational Analysis

Our complex structure of SaSsbA with 5-FU has elucidated the binding mode and identified the interactive residues. Notably, a substantial conformational change was observed in R18 upon 5-FU binding, resulting in a shift of its side chain position by 7.2 Å and an angular alteration of 77.4°. This suggests that R18 might play a role in initiating or mediating the 5-FU binding process. Since R18 is highly conserved among SSB homologs (Figure 1), we generated an alanine substitution mutant (Table 2), which was subsequently purified and analyzed to investigate its contribution to binding (Figure 6). The strength of interaction between the R18A mutant and 5-FU was assessed through fluorescence quenching and compared to that of the wild-type SaSsbA (WT). The quenching phenomenon involves the formation of a complex that diminishes the protein’s fluorescence intensity. SaSsbA exhibited prominent intrinsic fluorescence, with a peak wavelength at 339 nm upon excitation at 279 nm (Figure 6A). As concentrations of 5-FU were incrementally added to the SaSsbA solution, the intrinsic fluorescence underwent gradual quenching (Figure 6A). Upon introducing 200 μM of 5-FU, the intrinsic fluorescence of SaSsbA was reduced by 83.8% (Table 3). The binding of 5-FU induced a red shift of the SaSsbA emission wavelength from 339 nm to 347 nm (about 8 nm), as indicated by the change in *λ*_max_ (Table 3). These observations collectively confirm the formation of a stable complex between SaSsbA and 5-FU. Comparable to SaSsbA, the R18A mutant also exhibited strong intrinsic fluorescence, with a peak wavelength at 339 nm upon excitation at 279 nm (Figure 6B). However, the addition of 200 μM 5-FU resulted in only a 34.9% reduction in the intrinsic fluorescence of R18A (Table 3). Furthermore, the *λ*_max_ of R18A shifted only minimally from 339 nm to 340 nm upon exposure to 200 μM 5-FU. Analysis of the titration curves (Figure 6C) facilitated the determination of *K*_d_ values of 497.6 ± 13.5 μM for R18A and 55.9 ± 0.7 μM for WT (Table 3). These experimental findings observed from structural and functional investigations collectively underscore the significance of R18 in SaSsbA as a crucial residue for 5-FU binding.

### 2.12. Distinct 5-FU Binding Modes in SaSsbA and SaSsbB

Unlike the case of *E. coli*, which possesses a singular type of SSB (EcSSB), certain bacteria, particularly some Gram-positive bacteria [54], harbor two paralogous SSBs, namely SsbA and SsbB. Recently, we elucidated the crystal structure of SaSsbB [46], as well as its complex with 5-FU [49]. Consequently, the complex structure of SaSsbB is available for a comparative analysis of the 5-FU binding mode in relation to SaSsbA (PDB ID 7YM1) and SaSsbB (PDB ID 7D8J). Given the structural similarity between SaSsbA and SaSsbB, one might naturally assume a congruent 5-FU binding mode. However, intriguingly, our complexed structure revealed distinct 5-FU binding configurations for SaSsbA and SaSsbB (Figure 7). In the complexed structure, specific residues including T12, K13, T30, F48, and N50 of SaSsbB were identified to interact with 5-FU, forming an integral part of the binding site (Figure 7A) [49]. Particularly noteworthy is the essential stacking interaction between the pyrimidine ring of 5-FU and the aromatic ring of F48 in SaSsbB, which underpins the drug–protein interaction. While these residues are entirely conserved in SaSsbA, they did not engage in interactions with 5-FU (Figure 7A). Upon superposition analysis, a considerable distance of 18.8 Å was evident between these divergent 5-FU binding sites (Figure 7B). Furthermore, the residues in SaSsbA that interact with 5-FU are similarly conserved in SaSsbB; however, they do not collectively form a 5-FU binding site in SaSsbB (Figure 7C). Given the perfect conservation of these 5-FU binding residues in both SaSsbA and SaSsbB, the substantial dissimilarity observed warrants further investigation to ascertain whether other inherent species-specific differences contribute to this phenomenon. Additional biophysical studies are warranted to comprehensively explore these disparities.

## 3. Discussion

Metabolic reprogramming is the strategy adopted by cancer cells to expedite their proliferation, resist the effects of chemotherapy, invade tissues, metastasize, and endure within nutrient-scarce microenvironments [2]. Various uracil derivatives have long been harnessed as pyrimidine-based antimetabolites in the battle against cancer [63]. Chief among these agents is 5-FU [4], a prominent fluoropyrimidine drug esteemed for its role in targeting TSase during anticancer chemotherapy [9]. Over the past 60 years, chemotherapeutic agents designed to thwart thymidylate biosynthesis have emerged as stalwarts in cancer treatment. In addition, the synergistic administration of 5-FU alongside other chemotherapeutic agents amplifies treatment efficacy and overall survival rates, particularly in cancers involving the head, breast, and neck [64]. However, some mechanistic details, including signaling pathways, remain unexplained [65,66,67]. It is important to recognize that the purview of 5-FU’s interactions goes beyond merely engaging human TSase. For example, other human proteins, including dihydroorotase, PARP (procyclic acidic repetitive protein), VEGFR1 (vascular endothelial growth factor receptor 1), and CASP-3 (caspase-3 protein), are also known to interact with 5-FU [14,68]. Moreover, the intricate interplay between microbiota and chemotherapeutic drugs, such as 5-FU, holds the potential to influence host responses, further adding to the complexity of the landscape [17]. Consequently, a comprehensive elucidation of the complete 5-FU interactome is imperative, serving as the foundation for exhaustive clinical pharmacokinetic assessments and toxicity analyses [21,22]. The holistic elucidation of the multifaceted relationships woven by 5-FU holds the promise of enhancing our grasp of its intricate dynamics, paving the way for refined therapeutic strategies and personalized medicine.

In this study, our findings have unveiled SaSsbA’s capacity to engage in interaction with the anticancer drug 5-FU (Figure 6). In comparison with SaSsbB, a paralogous protein of SaSsbA in *S. aureus*, the *K*_d_ values for 5-FU binding to SaSsbA and SaSsbB are 55.9 ± 0.7 μM (Table 3) and 152.8 ± 2.5 μM [49], respectively. Based on the fact that the *K*_d_ value of human dihydroorotase bound to 5-FU is 91.2 ± 1.7 μM [14], the hierarchy of binding affinities for 5-FU can be delineated as follows: SaSsbA > human dihydroorotase > SaSsbB. This outcome may imply that, in scenarios where 5-FU enters the human system, it exhibits a preference for binding to the bacterial DNA replication protein SaSsbA within bacterial cells, as opposed to its interaction with the human enzyme dihydroorotase. However, it is important to underscore that this supposition necessitates comprehensive validation through a thorough investigation spanning biochemical and cellular dimensions. Considering the potential diversity of the gut microbiome across individuals, it remains imperative to ascertain the binding affinities of 5-FU to any feasible proteins within the human body, encompassing locales like the gastrointestinal tract and bloodstream. Such investigations are crucial to facilitate comprehensive comparisons and subsequent clinical analyses.

For the investigation of the binding mode, we solved the complexed crystal structure of SaSsbA with 5-FU at a resolution of 2.3 Å (Table 1). The interaction between 5-FU and SaSsbA was found to involve R18, P21, V52, F54, Q78, R80, E94, and V96 (Figure 4). Unexpectedly, this pattern of interactive residues deviated entirely from those identified in the 5-FU binding sites of SaSsbB [49]. In contrast to SaSsbA, where the interaction with 5-FU relies on a distinct set of residues, SaSsbB’s 5-FU interaction hinges on T12, K13, T30, F48, and N50 (Figure 7). Notably, the stacking interaction between the aromatic ring of F48 and the pyrimidine ring of 5-FU assumes a pivotal role in the drug–protein interaction within SaSsbB [49]. Intriguingly, despite the presence of these identical residues within both SaSsbA and SaSsbB, none of them (T12, K13, T30, F48, and N50) participate in 5-FU binding within SaSsbA. Biochemically, reconciling the divergence in 5-FU binding sites, despite the residue conservation between SaSsbA and SaSsbB, presents a challenge. Although the OB folds share a striking visual resemblance, we noted a subtle difference in the size of the binding groove between SaSsbA and SaSsbB. This structural divergence is underscored by the angles between strands β1’ and β4 in their monomers A, which measure 41.2° and 65.9°, respectively (Figure 5). This variance in binding groove width may potentially influence the mechanisms governing 5-FU binding. A noteworthy observation pertains to the substantial conformational alteration that accompanies 5-FU binding (Figure 5). This observation led us to propose a hypothesis wherein these seemingly identical residues result in divergent 5-FU binding modes. Within various contexts, OB folds can exhibit broad ligand-binding capabilities, targeting both single-stranded DNA and proteins [35]. This is evident in cases such as the tumor suppressor BRCA2, where two OB folds bind to ssDNA while a third participates in protein–protein interactions rather than ssDNA binding [69]. ssDNA bound by *Pseudomonas aeruginosa* SSB (PaSSB) only occupies half of the binding sites of two OB folds rather than four OB folds through the ssDNA-binding mode (SSB)_3:1_ [57,58]. Similarly, in RPA, two different binding modes involve two and four OB folds, respectively [70]. Insights gleaned from the SaSsbA-glycerol complex structure (Figure 3) indicate that Glycerol 1 and 2 do not necessarily need to occupy corresponding sites within monomers A and B. Previous experimental observations through single molecule experiments [71,72] also suggest that the unoccupied OB fold within SSB could adopt an open conformation to facilitate sliding, and, therefore, the ligand-binding groove within its OB-fold structure could be regulated to accommodate the requirements of dynamic binding processes (Figure 7). Consequently, it becomes plausible to consider that even when similar sites exist, 5-FU may bind to different locations due to the influence of these adaptable binding grooves.

Differing from some enzymes characterized by a single active site, the binding behavior of the DNA replication protein SaSsbA, which engages with diverse ssDNA and proteins, can introduce unpredictability into its ligand binding site(s). A multitude of solvent-exposed surfaces on SaSsbA functions as binding sites for both ssDNA and partner proteins, further complicating the task of foreseeing which specific pocket serves as the binding site. The prospect of utilizing docking tools, like MOE Dock [73], to anticipate a protein’s ligand binding site presents itself as a viable avenue to explore. The analysis conducted through MOE Dock highlighted five preferred binding modes (Figure 8). However, none of these predicted sites (Table 4) aligned precisely with the binding site evident in the complexed crystal structure of SaSsbA with 5-FU (Figure 2). Therefore, it becomes evident that the generation of additional complexed crystal structures remains an imperative in facilitating more robust binding analyses and supporting the development of structure-based approaches to drug design.

It is believed that all current cells trace their lineage back to a shared ancestor, suggesting that fundamental principles learned from experiments conducted with one cell type possess broad applicability across diverse cells. This perspective implies that the mechanisms governing essential cellular activities, such as DNA replication, transcription, and translation, should exhibit similarities across various cell types. Nonetheless, in response to challenging environmental circumstances, organisms tend to evolve new enzymes or auxiliary components to enhance their survival prospects and adaptive capabilities throughout evolutionary processes. In contrast to the situation in *E. coli* and many other bacteria, which feature a single SSB, certain microorganisms like *S. aureus* and other Gram-positive bacteria manifest multiple paralogous SSBs, including SsbA [74], SsbB [59], and SsbC [47]. Intriguingly, the positioning of the *ssbA* gene in the *S. aureus* genetic map does not align with the location of the *ssb* gene in *E. coli* and other Gram-negative bacteria [53]. Notably, this corresponding position in *E. coli* is occupied by *priB* [53], another variant of SSB [55,75]. Given the diversity of primosomal proteins with which these SSBs interact [76,77], SaSsbA and EcSSB confront an array of binding partners within their respective cellular contexts. This intricacy suggests that the presence of these distinct SSBs might necessitate their co-evolution with partner proteins, enabling the development of species-specific functions to address survival demands and secure a competitive edge. This co-evolutionary dynamic might elucidate the lack of conservation in PXXP motifs and amino acid residues within the IDL among different SSBs, including SaSsbA (Figure 1) [39]. Furthermore, intriguing disparities come to light, such as myricetin’s inhibitory effect on PaSSB but not on SaSsbA [44] or *Klebsiella pneumoniae* SSB [40]. Even within proteins that share structural similarities, as demonstrated by SaSsbA and SaSsbB, their 5-FU binding modes exhibit complete divergence (Figure 7). Consequently, it is conceivable that 5-FU could bind to these distinct SSBs present in both human and microorganisms, subsequently influencing various cellular signaling pathways. Despite these observations, additional research is indispensable to elucidate the precise mechanisms underpinning the recognition of 5-FU binding sites and the rationale behind the evolution of these diverse SSBs within specific species.

## 4. Materials and Methods

### 4.1. Protein Expression and Purification

The expression vector pET21b-SaSsbA [45] was transformed into *E. coli* BL21 (DE3) cells and grown in LB medium at 37 °C. The overexpression was induced by incubating with 1 mM isopropyl thiogalactopyranoside for 9 h. Recombinant SaSsbA was purified from the soluble supernatant by using Ni^2+^-affinity chromatography. The recombinant protein was eluted with a linear imidazole gradient and dialyzed against a dialysis buffer (20 mM Tris-HCl and 0.1 M NaCl, pH 7.9; Buffer A). Protein concentration was measured using Biorad protein (Bradford) assay. The protein purity remained at >97% as determined using SDS–PAGE.

### 4.2. Site-Directed Mutagenesis

The SaSsbA mutant was generated according to the QuikChange site-directed mutagenesis kit protocol (Stratagene; LaJolla, CA, USA), by using the wild-type plasmid pET21b–SaSsbA as a template. The presence of the mutation was verified by DNA sequencing. The recombinant mutant proteins were purified using the protocol for the wild-type SaSsbA by Ni^2+^-affinity chromatography.

### 4.3. Crystallization Experiments

Purified SaSsbA was concentrated to 20 mg/mL with addition of the cryoprotectant glycerol to a final concentration of 25% for storage at −20 °C. The crystals of the SaSsbA-5-FU complex appeared at room temperature through hanging drop vapor diffusion in 16% PEG 4000, 100 mM Tris-HCl, 200 μM 5-FU, and 200 mM MgCl_2_ at pH 8.5. For the SaSsbA–glycerol complex, the crystals were grown in 30% PEG 4000, 100 mM HEPES, 200 mM CaCl_2_ at pH 7.5. These crystals reached full size in 7–13 days and validated in the beamline 15A of the National Synchrotron Radiation Research Center (NSRRC; Hinchu, Taiwan).

### 4.4. X-ray Diffraction Data and Structure Determination

Data were collected in the beamline 15A using an Rayonix MX300HE CCD Area Detector at NSRRC. Data sets were indexed, integrated, and scaled using HKL-2000 [78] and XDS [79]. The initial phase, density modification, and model building were performed using the AutoSol program in the PHENIX [80]. The iterative model building and structure refinement were performed using Refmac in the CCP4 software suite (version v7.1.008) [81] and phenix.refine in the PHENIX software suite (Phenix1.19.1-4122) [82]. The initial phase of SaSsbA complexed with 5-FU was determined through the molecular replacement software Phaser MR (Phenix1.19.1-4122) [83] by using SaSsbA (PDB ID 5XGT) as a search model. The correctness of the stereochemistry of the models was verified using MolProbity [84].

### 4.5. Fluorescence Quenching

The *K*_d_ value of purified SaSsbA was determined using the fluorescence quenching method previously described for the DHOase [85] and DHPase [86,87]. Briefly, an aliquot of the compound was added into the solution containing SaSsbA (1 μM) and 50 mM HEPES at pH 7.0. SaSsbA displayed strong intrinsic fluorescence with a peak wavelength of 339 nm when excited at 279 nm at 25 °C. The decrease in the intrinsic fluorescence of SaSsbA was measured at 339 nm with a spectrofluorometer (Hitachi F-2700; Hitachi High-Technologies, Japan). The *K*_d_ was obtained using the following equation: Δ*F* = Δ*F*_max_ − *K*_d_(Δ*F*/[5-FU]). Data points are an average of 2–3 determinations within a 10% error.

### 4.6. MOE-Dock Analysis

The binding analysis of 5-FU to SaSsbA was carried out using MOE-Dock (version 2019.0102) [73]. The binding capacity was also calculated using MOE-Dock. The crystal structure of SaSsbA (PDB ID 5XGT) was used [45]. Before docking, any water molecules present in the crystal structure were removed using MOE. To ensure accuracy, a 3D protonation step followed by energy minimization was applied to add hydrogen atoms to the protein structure. The binding modes were generated and predicted through the MOE-Dock tool and visualized by PyMOL.

## 5. Conclusions

SsbA represents a captivating molecular apparatus that orchestrates a multitude of indispensable processes vital for maintaining DNA integrity [88]. This study has revealed SaSsbA’s hitherto unknown capability of binding to the anticancer drug 5-FU, thereby expanding the roster of proteins within the 5-FU interactome to encompass this pivotal DNA replication protein (Figure 6). In light of the results derived from mutational and structural analyses, it became evident that SaSsbA’s mode of binding with 5-FU diverges from that of SaSsbB (Figure 7). Given the insights offered by the glycerol and 5-FU binding sites (Figure 3 and Figure 4), our complexed SaSsbA structures underscore the likelihood that several of these interactive residues could be suitable targets for drug interventions aimed at inhibiting SaSsbA activity. This complexed structure also holds the potential to furnish valuable comprehension regarding how 5-FU and its pyrimidine derivatives might bind to and impede analogous OB-fold proteins in humans, particularly within cancer-related signaling pathways [89]. Acknowledging the capacity of microbiota to influence the host’s response to 5-FU, there emerges a pressing need for further research to revisit the roles that bacterial and human SSBs play in the realm of anticancer therapy.

## Figures and Tables

**Figure 1 ijms-24-14899-f001:**
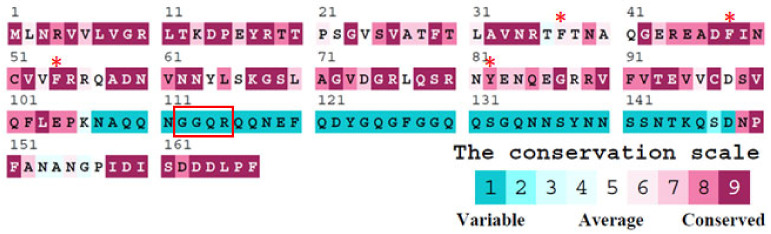
Sequence analysis of SaSsbA reveals an alignment consensus of sequenced SSB homologs through ConSurf analysis, which effectively showcases the extent of variability exhibited at each position along the sequence. In this depiction, amino acid residues that span a wide range of variability are depicted in teal, while those that remain highly conserved are marked in burgundy. Residues F37, F48, F54, and Y82, which potentially participate in ssDNA binding through stacking interactions, are denoted by asterisks. Of significance is the corresponding GGRQ motif found in SaSsbA, which is highlighted within a red box. The PXXP motifs seen in EcSSB are notably absent in SaSsbA. Another notable contrast arises in the critical C-terminal tail. While the DDDIPF sequence of EcSSB plays a vital role in protein–protein interaction, this sequence takes the form of DDDLPF in SaSsbA.

**Figure 2 ijms-24-14899-f002:**
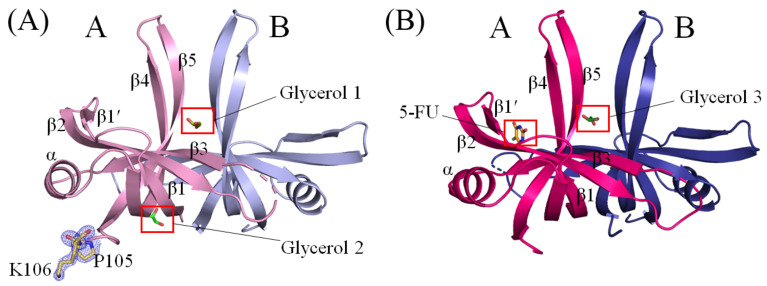
Crystal Structures of SaSsbA. (**A**) The crystal structure of SaSsbA featuring two glycerol molecules is displayed. The SaSsbA monomer adopts an OB-fold structure, composed of a β-barrel comprising 5 β-strands, capped with an α-helix. These two glycerol molecules are designated as Glycerol 1 (splitpea) and Glycerol 2 (green). Different SaSsbA monomers are colored in light pink and light blue. In contrast to the crystal structure of apo-SaSsbA, this complexed structure displayed two additional amino acid residues, namely P105 and K106, as indicated by the blue mesh. (**B**) The complexed crystal structure of SaSsbA with one 5-FU molecule and one glycerol molecule is presented. Different SaSsbA monomers are colored in hot pink and deep blue. Notably, 5-FU was localized solely in monomer A, with no presence in monomer B. The glycerol molecule is labeled as Glycerol 3 (forest).

**Figure 3 ijms-24-14899-f003:**
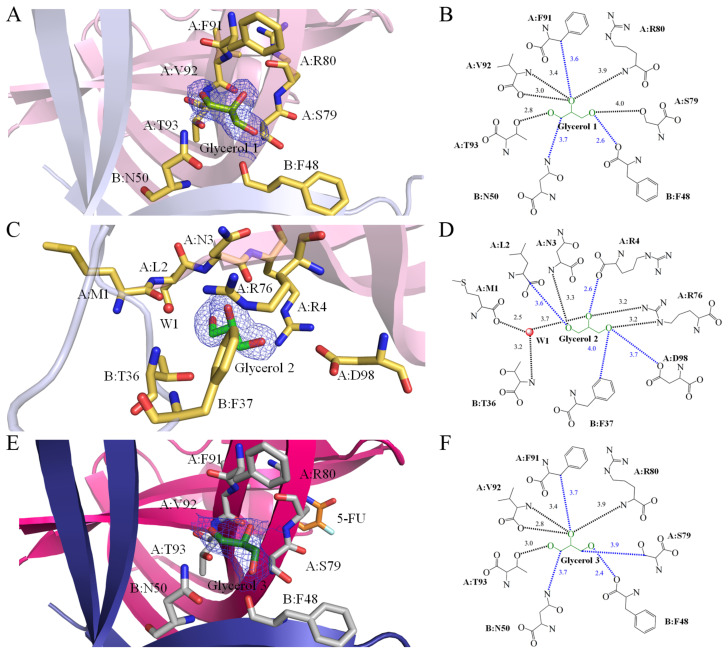
Glycerol binding modes. (**A**) The binding site of Glycerol 1 within SaSsbA was unveiled through the structure of glycerol-bound SaSsbA (PDB ID 8GW5). Glycerol 1 (splitpea) was positioned between SaSsbA monomers A (light pink) and B (light blue). Residues engaging with Glycerol 1 are colored in yellow. The composite omit map (presented as blue mesh, contoured at 1 σ) indicated the presence of Glycerol 1 within a cavity formed at the SaSsbA monomers A and B interface. (**B**) Depiction of the binding mode for Glycerol 1. Residues F48 (Subunit B), N50 (Subunit B), S79 (Subunit A), R80 (Subunit A), F91 (Subunit A), V92 (Subunit A), and T93 (Subunit A), situated within a contact distance of <4 Å, were instrumental in binding Glycerol 1. The interactive distances are also shown in Å. Based on interactions detected by PLIP, hydrogen bonds were formed between the main chains of R80 and V92, as well as the side chains of S79 and T93, and Glycerol 1 (indicated in black). (**C**) The binding site of Glycerol 2 within SaSsbA was unveiled by the structure of glycerol-bound SaSsbA (PDB ID 8GW5). Similar to Glycerol 1, Glycerol 2 (green) interacted with both SaSsbA monomers A (light pink) and B (light blue). However, the binding poses and locations between Glycerol 1 and Glycerol 2 exhibited distinctions. (**D**) Depiction of the binding mode for Glycerol 2. Within a contact distance of <4 Å, M1 (Subunit A), L2 (Subunit A), N3 (Subunit A), R4 (Subunit A), T36 (Subunit B), F37 (Subunit B), R76 (Subunit A), and D98 (Subunit A) were engaged in binding Glycerol 2. Structural analysis via PLIP revealed hydrogen bonds formed between the main chains of M1, N3, and T36, as well as the side chain of R76, and Glycerol 2. (**E**) The binding site of Glycerol 3 within SaSsbA was revealed by the SaSsbA–5-FU complex structure (PDB ID 7YM1). Glycerol 3 (forest) was nestled between SaSsbA monomers A (hot pink) and B (deep blue). The binding mode for Glycerol 3 exhibited similarities to that of Glycerol 1. 5-FU (orange) was also showcased within this structure. (**F**) Depiction of the binding mode for Glycerol 3. Through interactions detected using PLIP, it was determined that hydrogen bonds were formed between the main chains of R80 and V92, along with the side chain of T93, and Glycerol 3.

**Figure 4 ijms-24-14899-f004:**
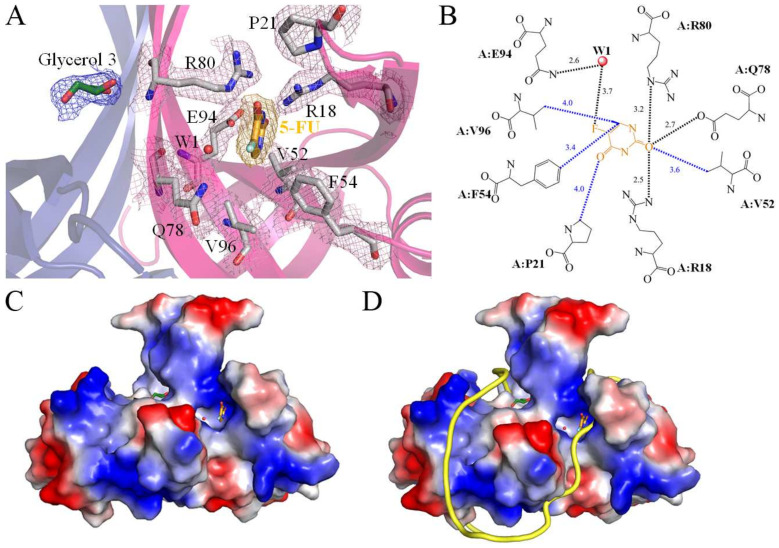
5-FU interaction mode. (**A**) The binding site for 5-FU within SaSsbA was unveiled through the SaSsbA–5-FU complex structure (PDB ID 7YM1). This complexed structure of an SaSsbA dimer contained one glycerol molecule (Glycerol 3; forest) and one 5-FU molecule (orange). The residues engaged in interactions with 5-FU are depicted in gray. An orange mesh, contoured at 1 σ, illustrates the presence of 5-FU within the groove of SaSsbA monomer A (hot pink). The electron density for these interactive residues is also distinctly visible (light pink mesh, contoured at 1 σ). (**B**) Depiction of the binding mode for 5-FU. Residues R18, P21, V52, F54, Q78, R80, E94, and V96, situated within a contact distance of <4 Å, played pivotal roles in binding with 5-FU. The corresponding interactive distances are also indicated (Å). Based on the interactions identified via PLIP, the side chains of R18, Q78, R80, and E94 engaged in hydrogen bonding with 5-FU (highlighted in black). (**C**) The electrostatic potential surface portrayal of the SaSsbA complexed with 5-FU elucidates the distribution of positive (blue) and negative (red) charges. Notably, 5-FU (orange) occupies a groove within SaSsbA, potentially pertinent to ssDNA binding. (**D**) The superimposed structures of the SaSsbA–5-FU complex and the EcSSB–ssDNA complex (PDB ID 1EYG). The crystal structures of SaSsbA and EcSSB exhibit similarity. For clarity, the EcSSB structure is omitted. The distribution of positive (blue) and negative (red) charges showcases a collection of fundamental basic residues on the surface of SaSsbA, which are exposed to the solvent and collectively form a binding pathway conducive for accommodating ssDNA binding (gold). Our complex structure highlights that the 5-FU presence within the groove may potentially affect and regulate the ssDNA wrapping phenomenon by SaSsbA.

**Figure 5 ijms-24-14899-f005:**
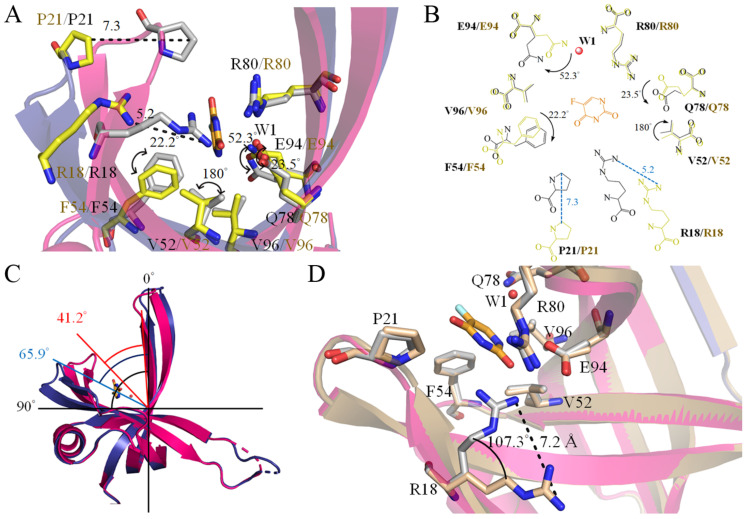
Comparative analysis of 5-FU binding sites in different states of SaSsbA. (**A**) Superposition of monomer A (hot pink) and monomer B (deep blue) within the 5-FU complexed SaSsbA structure (PDB ID 7YM1). Residues in monomer A and B are depicted in gray and yellow, respectively, with the presence of a 5-FU molecule shown in orange. (**B**) Comparison of the 5-FU binding sites between the 5-FU-bound and glycerol 3-bound states of SaSsbA. Binding of 5-FU led to spatial shifts of R18 and P21 by distances of 5.2 and 7.3 Å, respectively. Additionally, V52, F54, Q78, and E94 underwent angular shifts of 180, 22, 23, and 52°, respectively, upon 5-FU binding. (**C**) Evaluation of the sizes of the binding groove in monomer A and B. Despite the comparable appearance of their OB folds, variations in their sizes were noted. Notably, the structural angles between strands β1′ and β4 in monomer A and B were found to be 41.2 and 65.9°, respectively. (**D**) Superposition of the 5-FU-bound (hot pink; PDB ID 7YM1) and unbound (pale yellow; PDB ID 5XGT) states of SaSsbA. Residues within the 5-FU-bound and unbound states are presented in gray and wheat hues, respectively. A 5-FU molecule is visualized in orange. Remarkably, the side chain of R18 experienced a considerable shift, spanning a distance of 7.2 Å and an angular shift of 107.3° upon 5-FU binding.

**Figure 6 ijms-24-14899-f006:**
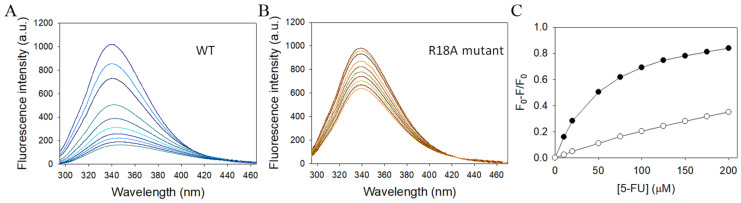
Fluorescence titration of SaSsbA with 5-FU. (**A**) The fluorescence emission spectra of WT with 5-FU of different concentrations (0–200 μM; 0, 10, 20, 50, 75, 100, 125, 150, 175, and 200 μM). The decrease in intrinsic fluorescence of protein was measured at 339 nm upon excitation at 279 nm with a spectrofluorometer. The fluorescence intensity emission spectra of SaSsbA were significantly quenched by 5-FU. (**B**) The fluorescence emission spectra of the mutant R18A with 5-FU of different concentrations (0–200 μM). The strength of interaction between the R18A mutant and 5-FU was assessed and compared to that of WT. (**C**) The titration curves for determining the *K*_d_ values of 5-FU for WT (●) and R18A (○). The *K*_d_ was obtained by the equation: Δ*F* = Δ*F*_max_ − *K*_d_(Δ*F*/[5-FU]). Data points are an average of 2–3 determinations within a 10% error.

**Figure 7 ijms-24-14899-f007:**
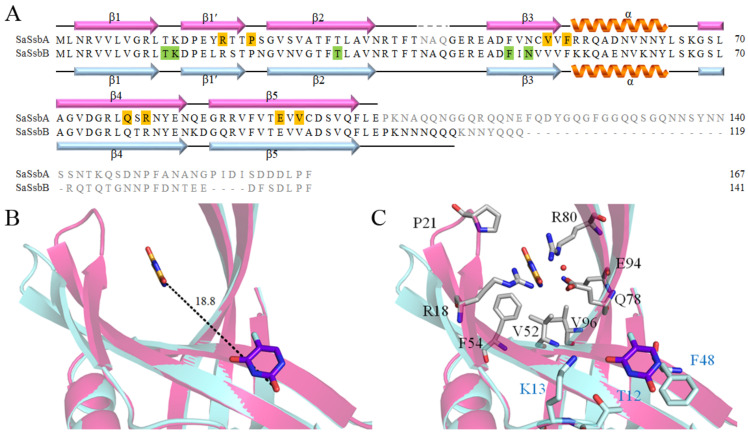
Distinct 5-FU binding modes in SaSsbA and SaSsbB. (**A**) Alignment of the sequences of SaSsbA and SaSsbB, with secondary structural elements indicated. Unobserved amino acids in these crystal structures are colored in gray. The residues responsible for 5-FU binding in SaSsbA and SaSsbB are shaded in orange and green, respectively. (**B**) Superposition of the 5-FU-bound structures of SaSsbA (hot pink) and SaSsbB (aquamarine). The 5-FU molecules in SaSsbA (PDB ID 7YM1) and SaSsbB (PDB ID 7D8J) are represented in orange and purple-blue, respectively. The distance between these distinct 5-FU binding sites measures 18.8 Å. (**C**) Disparate 5-FU binding sites. Despite the perfect conservation of these 5-FU binding residues in SaSsbA (gray) and SaSsbB (aquamarine), each protein exhibits a preferred binding site for 5-FU.

**Figure 8 ijms-24-14899-f008:**
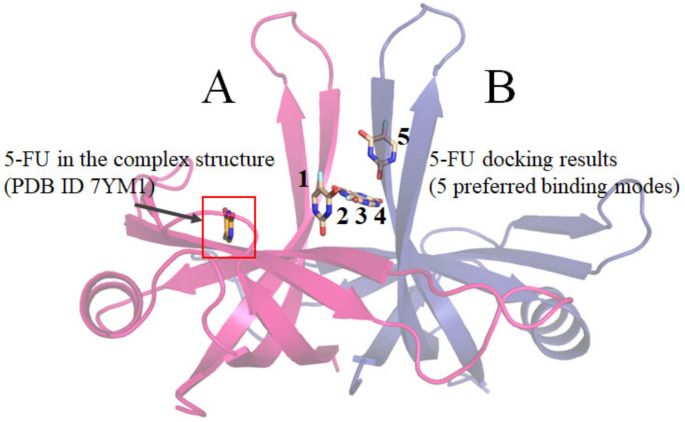
Molecular docking of 5-FU with SaSsbA. Comparison between 5-FU molecules extracted from the complexed SaSsbA structure and the outcomes of docking simulations. SaSsbA monomers A and B are colored in light pink and light blue. Molecular docking of 5-FU was performed using the crystal structure of the apo-form SaSsbA (PDB ID 5XGT). The binding affinities of SaSsbA with 5-FU, considering all feasible binding orientations, were evaluated using the S score. Utilizing MOE-Dock (version 2019.0102) software, we identified and numbered the five most favorable binding modes for 5-FU. The five preferred binding modes of 5-FU are indicated as 1–5. Additionally, the position of 5-FU in the complexed crystal structure of SaSsbA (PDB ID 7YM1) was included for reference.

**Table 1 ijms-24-14899-t001:** Data collection and refinement statistics.

Data Collection		
Crystal	SaSsbA–5-FU complex	Glycerol-bound SaSsbA
Wavelength (Å)	1	1
Resolution (Å)	27.8–2.36	27.9–1.80
Space group	P4_1_2_1_2	P4_1_2_1_2
Cell dimension *a*, *b*, *c* (Å) *β* (°)	88.09, 88.09, 57.78 90.00	88.22, 88.22, 58.00 90.00
Redundancy	7.8 (7.9)	9.4 (9.4)
Completeness (%)	99.9 (100.0)	100.0 (100.0)
<I/σI>	31.6 (8.2)	35.7 (4.4)
CC_1/2_	0.997 (0.974)	0.989 (0.955)
Refinement		
No. of reflections	9797	21,823
*R*_work_/*R*_free_	0.203/0.244	0.199/0.224
No. of atoms		
Protein	199	200
Water	52	151
5-FU	1	0
Glycerol	1	2
r.m.s deviations		
Bond lengths (Å)	0.008	0.007
Bond angles (°)	1.06	1.05
Ramachandran plot		
Favored (%)	98.43	97.40
Allowed (%)	1.57	2.60
Outliers (%)	0	0
PDB entry	7YM1	8GW5

Values in parentheses are for the highest resolution shell. CC_1/2_ is the percentage of correlation between intensities of random half-data sets.

**Table 2 ijms-24-14899-t002:** Primers used for construction of the plasmid.

Oligonucleotide	Primer
SaSsbA-R18A-N	GAAAGATCCGGAATACGCAACCACTCCCTC
SaSsbA-R18A-C	ACACCTGAGGGAGTGGTTGCGTATTCCGGA

Underlined nucleotides indicate the designated site for mutation site.

**Table 3 ijms-24-14899-t003:** Binding parameters of SaSsbA WT and the mutant R18A.

SaSsbA	*λ*_max_ (nm)	*λ*_em_ Shift (nm)	Quenching (%)	*K*_d_ Value (µM)
SaSsbA WT	339	347	83.81	55.9 ± 0.7
SaSsbA-R18A	339	340	34.99	497.6 ± 13.5

The decrease in the intrinsic fluorescence of SaSsbA was measured with a spectrofluorometer (Hitachi F-2700; Hitachi High-Technologies, Tokyo, Japan). The *K*_d_ was obtained using the following equation: Δ*F* = Δ*F*_max_ − *K*_d_(Δ*F*/[5-FU]).

**Table 4 ijms-24-14899-t004:** Molecular docking analysis.

Mode	S Score	Receptor Residue	Interaction	Distance (Å)	E (kcal/mol)
1	−4.2104	Phe 48 (A)	H-donor	2.96	−5.6
		Thr 93 (A)	H-donor	2.98	−1.9
		Asn 50 (A)	H-acceptor	3.08	−1.1
2	−4.1638	Arg 80 (A)	H-donor	3.42	−0.9
		Val 92 (A)	H-donor	3.03	−1.2
		Asn 81 (A)	H-acceptor	3.14	−2.2
3	−3.9845	Arg 80 (A)	H-donor	3.27	−1.0
		Val 92 (A)	H-donor	3.05	−1.2
		Phe 91 (A)	Pi-H	4.03	−0.6
4	−3.8762	Asn 50 (B)	H-donor	3.37	−1.8
		Asn 81 (A)	H-acceptor	3.52	−0.8
5	−3.8744	Phe 91 (A)	H-donor	3.07	−1.6
		Phe 91 (A)	H-acceptor	3.15	−1.2

## Data Availability

Atomic coordinates and related structure factors were deposited in the PDB with accession codes 7YM1 and 8GW5.

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
