# Peer review of "Crystal Structure of DNA Replication Protein SsbA Complexed with the Anticancer Drug 5-Fluorouracil"

_ijms, 2023, doi:10.3390/ijms241914899_

Round 1

Reviewer 1 Report

The manuscript titled “Crystal structure of DNA replication protein SsbA complexed with the anticancer drug 5-fluorouracil reveals a novel binding mode and extends the structural interactome of 5-FU by Su et al., characterize the binding of 5-Fluorouracil to Ssba Staphylococcus aureus using X-ray crystallography and fluorescence spectroscopy. Importance of particular residue is also determined by Site-directed mutagenesis. As authors already published Ssbb Structure before and It seems like this work likely the extension of the earlier work. The present manuscript will add some insights to the presently available data on the small molecule- Single-Stranded binding protein binding characteristics. However, the present Manuscript is written poorly with less impressive outputs. The authors need to focus more on the presentation of figures and text. The Discussion section needs to be shortened and more specific and the conclusion section is like a vague section.  Needs to work on the English language. There were many typo errors throughout the manuscript. 

The Title can also rephrase it somewhat to make it shorter

Authors need to select either Three-letter or one-letter code for amino acid representations to make uniformity throughout the manuscript.

The glycerol is an additive or purification component? How does glycerol come in the binding?

The methodology needs to be described properly in detail.

 The fluorescence spectroscopy section include separately in the method section.

How was protein concentration measured? 

Statistical analysis in biochemical experiments? 

Overall, the present manuscript in the current version is not considerable for publication in IJMS.

Text English needs to improve.

Author Response

The manuscript titled “Crystal structure of DNA replication protein SsbA complexed with the anticancer drug 5-fluorouracil reveals a novel binding mode and extends the structural interactome of 5-FU by Su et al., characterize the binding of 5-Fluorouracil to Ssba Staphylococcus aureus using X-ray crystallography and fluorescence spectroscopy. Importance of particular residue is also determined by Site-directed mutagenesis.

Res: We thank the reviewer for the positive comments. Thank you very much.

As authors already published Ssbb Structure before and It seems like this work likely the extension of the earlier work. The present manuscript will add some insights to the presently available data on the small molecule- Single-Stranded binding protein binding characteristics.

Res: We thank the reviewer for the positive comments. We found that they are similar proteins but with totally different 5-FU binding modes. Thank you very much.

However, the present Manuscript is written poorly with less impressive outputs. The authors need to focus more on the presentation of figures and text.

Res: We have provided and used a new figure 1 with higher resolution accordingly. Thank you very much.

The Discussion section needs to be shortened and more specific and the conclusion section is like a vague section.

Res: The referee 2 commented “This is a very nice work showing at atomic resolution how the 5-FU drugs show two different binding modes on the SaSsbs giving a solid comprehension for futures drug development on this key participant of the DNA replication.” However, according to your valued comment, we still have shortened the Discussion section accordingly. We have also modified the Conclusion section accordingly. Thank you very much.

Needs to work on the English language. There were many typo errors throughout the manuscript.

Res: The referee 2 commented “English language fine. No issues detected.” According to your valued comment, we have found a typo and fixed it accordingly. Thank you very much.

The Title can also rephrase it somewhat to make it shorter

Res: We have made it shorter accordingly. The resultant title is “Crystal structure of DNA replication protein SsbA complexed with the anticancer drug 5-fluorouracil”. Thank you very much.

Authors need to select either Three-letter or one-letter code for amino acid representations to make uniformity throughout the manuscript.

Res: One-letter code for amino acid representations were used throughout the manuscript accordingly. Thank you very much.

The glycerol is an additive or purification component? How does glycerol come in the binding?

Res: The glycerol is an additive. It is common to find it in many other proteins being solved by X-ray crystallography, also as indicated by the referee 2. We have modified “crystal structure of SaSsbA complexed with glycerol” to Crystal Structure of glycerol-bound SaSsbA throughout the manuscript accordingly. Thank you very much.

The methodology needs to be described properly in detail.

Res: We have added the protein concentration (20 mg/mL) used for crystallization accordingly. JBScreen Classic 2 was found useful to find the crystallization conditions. We have added this information accordingly. Thank you very much.

The fluorescence spectroscopy section include separately in the method section.

Res: We have included Fluorescence Quenching section instead of determination of Kd accordingly. Thank you very much.

How was protein concentration measured?

Res: By using Biorad Protein (Bradford) Assay. We have added this information accordingly. Thank you very much.

Statistical analysis in biochemical experiments?

Res: The Kd was obtained by the equation: ΔF = ΔFmaxKdF/[5-FU]). Data points are an average of 2–3 determinations within a 10% error. We have added this information accordingly. Thank you very much.

Overall, the present manuscript in the current version is not considerable for publication in IJMS.

Res: In this study, we have identified that SsbA is a novel 5-FU binding protein. Further, its binding mode was confirmed and revealed by the crystal structure. This binding mode of SsbA is totally different from that of SsbB. This study extends the structural interactome of 5-FU. Thank you very much.

Reviewer 2 Report

Thank you for sending me this interesting manuscript by Hsin-Hui Su, et. al.  In this study, authors identified Staphylococcus aureus SsbA (SaSsbA) as a new participant of the anticancer drug 5-fluorouracil (5-FU) binding proteins. Hsin-Hui Su, et. al. solved the complexed crystal structure with 5-FU at a resolution of 2.3 Å. The structure of SaSsbA complexed with glycerol was also determined at 1.8 Å.

This is a very nice work showing at atomic resolution how the 5-FU drugs show two different binding modes on the SaSsbs giving a solid comprehension for futures drug development on this key participant of the DNA replication.

There are minor issues that are listed in order, as follows:

1) Figure 1, please provide a figure at higher resolution

2) Page 4, line 172: “Efforts were initially directed towards soaking and cocrystallization of SaSsbA with 5-FU (200 μM)”, here it is mentioned the 5-FU concentration, but it is not described the protein concentration, please add that information.

3) page 4, line 175:  “Subsequent rescreening was undertaken utilizing commercial crystallization kits”, please indicate the commercial crystallization kit name and what company was purchased from and mention if all the different conditions from that or those kits were tested.

4) Figure 2B: “The complexd crystal structure of SaSsbA with one 5-FU molecule and one glycerol molecule is presented” please fix the typo error.

5) Figure 2, please describe the structure in the model that appears within the blue mesh (at the bottom, panel A). It it’s not clear why the glycerol in the panel B is name as glycerol 3 when it is the only glycerol is found in that data set.

6) is it valid to say that SaSsbA was complexed to glycerol, if so, what roll is playing. Is it a ligand of SaSsbA. In my opinion it is the simple SaSsbA apo form, since the glycerol can be found not only in this crystal.Iit is common to find it in many other proteins being solved by X-ray crystallography.

Author Response

Thank you for sending me this interesting manuscript by Hsin-Hui Su, et. al. In this study, authors identified Staphylococcus aureus SsbA (SaSsbA) as a new participant of the anticancer drug 5-fluorouracil (5-FU) binding proteins. Hsin-Hui Su, et. al. solved the complexed crystal structure with 5-FU at a resolution of 2.3 Å. The structure of SaSsbA complexed with glycerol was also determined at 1.8 Å.

Res: We thank the reviewer for the positive comments. Thank you very much.

This is a very nice work showing at atomic resolution how the 5-FU drugs show two different binding modes on the SaSsbs giving a solid comprehension for futures drug development on this key participant of the DNA replication.

Res: We thank the reviewer for the positive comments. Thank you very much.

There are minor issues that are listed in order, as follows:

1) Figure 1, please provide a figure at higher resolution

Res: We have provided and used a new figure with higher resolution accordingly. Thank you very much.

2) Page 4, line 172: “Efforts were initially directed towards soaking and cocrystallization of SaSsbA with 5-FU (200 μM)”, here it is mentioned the 5-FU concentration, but it is not described the protein concentration, please add that information.

Res: 20 mg/mL. We have added this information accordingly. Thank you very much.

3) page 4, line 175: “Subsequent rescreening was undertaken utilizing commercial crystallization kits”, please indicate the commercial crystallization kit name and what company was purchased from and mention if all the different conditions from that or those kits were tested.

Res: The used commercial crystallization kits were purchased from Hampton Research and Jena Bioscience. JBScreen Classic 2 was found useful to find the crystallization conditions. We have added this information accordingly. Thank you very much.

4) Figure 2B: “The complexd crystal structure of SaSsbA with one 5-FU molecule and one glycerol molecule is presented” please fix the typo error.

Res: Complexed. We have fixed this typo error accordingly. Thank you very much.

5) Figure 2, please describe the structure in the model that appears within the blue mesh (at the bottom, panel A). It it’s not clear why the glycerol in the panel B is name as glycerol 3 when it is the only glycerol is found in that data set.

Res: We have added “In contrast to the crystal structure of apo-SaSsbA, this complexed structure displayed two additional amino acid residues, namely P105 and K106, as indicated by the blue mesh.” accordingly for describing the blue mesh.

The glycerol is only glycerol in this structure. To distinguish the position differences of the two glycerols in the panel A, we designated this one as Glycerol 3 for clarity.

Thank you very much.

6) is it valid to say that SaSsbA was complexed to glycerol, if so, what roll is playing. Is it a ligand of SaSsbA. In my opinion it is the simple SaSsbA apo form, since the glycerol can be found not only in this crystal.Iit is common to find it in many other proteins being solved by X-ray crystallography.

Res: We have modified “crystal structure of SaSsbA complexed with glycerol” to Crystal Structure of glycerol-bound SaSsbA throughout the manuscript accordingly. Thank you very much.

Round 2

Reviewer 1 Report

The authors incorporated the suggested comments and now agreed with MS acceptance.